# GUIDE: GUIDANCE-BASED INCREMENTAL LEARNING WITH DIFFUSION MODELS

## ABSTRACT

We introduce GUIDE, a novel continual learning approach that directs diffusion models to rehearse samples at risk of being forgotten. Existing generative strategies combat catastrophic forgetting by randomly sampling rehearsal examples from a generative model. Such an approach contradicts buffer-based approaches where sampling strategy plays an important role. We propose to bridge this gap by incorporating classifier guidance into the diffusion process to produce rehearsal examples specifically targeting information forgotten by a continuously trained model. This approach enables the generation of samples from preceding task distributions, which are more likely to be misclassified in the context of recently encountered classes. Our experimental results show that GUIDE significantly reduces catastrophic forgetting, outperforming conventional random sampling approaches and surpassing recent state-of-the-art methods in continual learning with generative replay.

## 1 INTRODUCTION

A typical machine learning pipeline involves training a model on a static dataset and deploying it to a task with a similar data distribution. This assumption frequently proves impractical in real-world scenarios, where models encounter a constantly evolving set of objectives. To address this issue, Continual Learning (CL) methods try to accumulate knowledge from separate tasks while overcoming catastrophic forgetting (French, 1999).

Typically, CL approaches can be divided into two main scenarios. In the *online* one, data arrives at the system in a continuous stream, and each example can be used for training only once. On the contrary, in the *offline* continual learning scenario, samples arrive in so-called tasks, allowing the model to run multiple epochs over a particular dataset before moving to the next one. To facilitate online continual learning, numerous methods (Rebuffi et al., 2017; Riemer et al., 2019; Benjamin et al., 2019; Aljundi et al., 2019b; Buzzega et al., 2020) utilize a memory buffer that stores a subset of training examples in order to replay them with every training batch. In order for this approach to be effective, it requires a well-designed sampling strategy that selects samples approximating the true loss on the previous task as closely as possible. Since the optimal selection of rehearsal samples is not possible, as it requires access to the classifier's

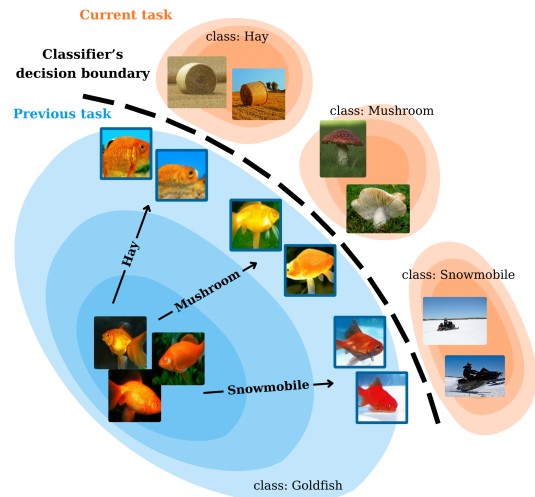

Figure 1: **Rehearsal sampling in GUIDE.** We guide the denoising process of a diffusion model trained on the previous task (**blue**) toward classes from the current task (**orange**). The replay samples, highlighted with **blue borders**, share features with the examples from the current task, which may be related to characteristics such as color or background (e.g., fishes on a snowy background when guided to *snowmobile*). Generative rehearsal on such samples positioned near the classifier's decision boundary successfully mitigates catastrophic forgetting.

future state (Khan & Swaroop, 2021), numerous methods try to achieve this goal by employing different heuristics. Recent approaches include selection based on the distance to the current decision boundary (Pan et al., 2020), maximization of the diversity of samples in a buffer (Aljundi et al., 2019b), loss increases given the estimated parameters update based on the newly arrived data (Aljundi et al., 2019a) or hard negative mining (Jin et al., 2021).

In this work, we propose to solve this problem with the Deep Generative Replay (DGR) (Shin et al., 2017) strategy, where the memory buffer is substituted with the generative model. In particular, we show that by first learning the whole data distribution of the previous task with Denoising Deep Probabilistic Models (DDPMs) (Sohl-Dickstein et al., 2015; Ho et al., 2020), we can use the classifier guidance technique (Dhariwal & Nichol, 2021) to steer the generation of rehearsal samples towards examples with high cross-entropy loss value, located close to the decision boundary at the given time of the classifier's continual learning. The visualization of this idea is presented in Fig. 1.

In our experiments, we show that rehearsal with GUIDE outperforms other state-of-the-art generative replay methods, significantly reducing catastrophic forgetting in class-incremental learning. On top of our method, we thoroughly evaluate several alternative guidance strategies that generate rehearsal samples of diverse characteristics. Our contributions can be summarized as follows:

- We introduce GUIDE - generative replay method that benefits from classifier guidance to generate rehearsal data samples prone to be forgotten.
- We demonstrate that incorporating classifier guidance enables the generation of high-quality samples situated near task decision boundaries. This approach effectively mitigates forgetting in class-incremental learning.
- We show the superiority of GUIDE over recent state-of-the-art generative rehearsal approaches and provide an in-depth experimental analysis of our method's main contribution.

## 2 RELATED WORK

### 2.1 GUIDED IMAGE GENERATION IN DIFFUSION MODELS

Besides conditioning, controlling diffusion model outputs can be achieved by modifying the process of sampling that incorporates additional signals from the guidance function. Classifier guidance (Dhariwal & Nichol, 2021) enables steering the backward diffusion process by combining the intermediate denoising steps of conditional or unconditional diffusion with the gradient from the externally trained classifier. To ensure the high quality of generated samples, the original method was introduced with a classifier trained on noised images. However, such an approach can be impractical and lead to limited performance.

Building upon the work by Bansal et al. (2023), in our method, we adapt the guidance process to utilize a classifier trained only on clean images. To that end, in the guidance process, we first predict a denoised image:

$$\hat{\mathbf{z}}_0(\mathbf{x}_t) = \frac{\mathbf{x}_t - \sqrt{1 - \bar{\alpha}_t}\epsilon_\theta(\mathbf{x}_t, t)}{\sqrt{\bar{\alpha}_t}}, \tag{1}$$

and then we modify the prediction of diffusion model $\epsilon_\theta(\mathbf{x}_t, t)$ at each time step $t$ according to:

$$\hat{\epsilon}_\theta(\mathbf{x}_t, t) = \epsilon_\theta(\mathbf{x}_t, t) + s\nabla_{\mathbf{x}_t}\ell\left(f_\phi(y|\hat{\mathbf{z}}_0(\mathbf{x}_t)), y\right), \tag{2}$$

where $s$ is gradient scale, $f_\phi(y|\mathbf{x})$ is classifier model with parameters $\phi$, $\ell$ is the cross-entropy loss function and $y$ is class label that we guide to.

In contrast to methods utilizing a classifier, Epstein et al. (2023) introduce self-guidance based on the internal representations of the diffusion model, while Ho & Salimans (2022) introduce classifier-free guidance, achieving results akin to classifier-based approaches by joint training of unconditional and conditional diffusion models.

### 2.2 CONTINUAL LEARNING

Continual learning methods aim to mitigate catastrophic forgetting – a phenomenon where deep neural networks trained on a sequence of tasks completely and abruptly forget previously learned

information upon retraining on a new task. Recent methods can be organized into three main families. **Regularization** methods (Kirkpatrick et al., 2017; Zenke et al., 2017; Li & Hoiem, 2017) identify the most important parameters and try to slow down their changes through regularization. **Architectural** approaches (Rusu et al., 2016; Yoon et al., 2018; Mallya & Lazebnik, 2018; Mallya et al., 2018; Verma et al., 2021) change the structure of the model for each task. **Rehearsal** methods replay data samples from previous tasks and train the model on a combination of data samples from previous and current tasks. In the most straightforward rehearsal approach, a memory buffer is used to store exemplars from previous tasks (Prabhu et al., 2020; Rebuffi et al., 2017; Chaudhry et al., 2018; Wu et al., 2019; Hou et al., 2019; Belouadah & Popescu, 2019; Castro et al., 2018; Aljundi et al., 2019b). Some methods, instead of directly using exemplars, stores data representations from previous tasks in different forms, e.g., *mnemonics* - optimized artificial samples (Liu et al., 2020b), distilled datasets (Wang et al., 2018; Zhao & Bilen, 2021; Zhao et al., 2021), or addressable memory structure (Deng & Russakovsky, 2022).

**Continual learning with generative rehearsal**   Because of the limitations of buffer-based approaches related to the constantly growing memory requirements and privacy issues, Shin et al. (2017) introduces Deep Generative Replay, where a GAN is used to generate rehearsal samples from previous tasks for the continual training of a classifier. A similar approach is further extended to different model architectures like Variational Autoencoders (VAE) (van de Ven & Tolias, 2018; Nguyen et al., 2018), normalizing flows (Scardapane et al., 2020) or Gaussian Mixture Models (Rostami et al., 2019).

On top of those baseline approaches, Ramapuram et al. (2020) introduce a method that benefits from the knowledge distillation technique in VAE training in CL setup, while Wu et al. (2018) introduce Memory Replay GANs (MeRGANs) and describe two approaches to prevent forgetting - by joint retraining and by aligning the replay samples. Instead of replaying data samples, several approaches propose to rehearse internal data representations instead, e.g., Brain-Inspired Replay (BIR) (Van de Ven et al., 2020) with an extension to Generative Feature Replay (GFR) (Liu et al., 2020a), where the rehearsal is combined with features distillation. Kemker & Kanan (2018) divide feature rehearsal into short and long-term parts.

**Continual learning with diffusion models**   Diffusion models excel in generative tasks, surpassing VAEs (Kingma & Welling, 2014) and GANs, yet their adoption in CL remains limited. Deep Diffusion-based Generative Replay (DDGR) (Gao & Liu, 2023) uses a diffusion model in a generative rehearsal method and benefits from a classifier pretrained on previous tasks to synthesize high-quality replay samples. Class-Prototype Conditional Diffusion Model (CDPM) (Doan et al., 2023) further enhances the replay samples quality by conditioning the diffusion model on learnable class-prototypes. Similarly, Jodelet et al. (2023) use an externally trained text-to-image diffusion model for the same purpose. In this work, we extend those ideas and show that classifier guidance might be used not only to generate high-quality images but also to introduce the desired characteristics of rehearsal samples.

## 3 METHOD

This section introduces GUIDE - a novel method designed to mitigate catastrophic forgetting in a classifier trained in a generative replay scenario with a diffusion model. We benefit from the classifier trained on the current task, referred to as the *current classifier*, to guide the diffusion model during the generation of rehearsal examples from preceding tasks. With our method, we can generate rehearsal examples close to the classifier's decision boundary, making them highly valuable to counteract the classifier's forgetting in class-incremental learning (Toneva et al., 2018; Kumari et al., 2022).

### 3.1 INTUITION AND RATIONALE BEHIND GUIDE

Before moving to the continual-learning setup, we demonstrate the effect of guiding the diffusion model towards classes not included in its training dataset. To that end, we propose a simplified scenario in which we employ the unconditional diffusion model $\epsilon_\theta(\mathbf{x}_t, t)$ trained exclusively on the *goldfish* and *tiger shark* classes from the ImageNet100 dataset, along with a classifier $f_\phi(y|\mathbf{x})$ trained on entire ImageNet dataset.

Given the diffusion model's unconditional nature, we use the classifier to steer the denoising process toward either the goldfish or tiger shark class, represented as $c_1$. Simultaneously, we add another guiding signal from the same classifier $f_\phi(y|\mathbf{x})$ towards one of the classes from the ImageNet dataset different than goldfish and tiger shark, denoted as $c_2$. Formally, we modify the unconditional diffusion sampling process as follows:

$$\hat{\epsilon}_\theta(\mathbf{x}_t, t) = \epsilon_\theta(\mathbf{x}_t, t) + s_1 \nabla_{\mathbf{x}_t} \ell\left(f_\phi(y|\hat{\mathbf{z}}_0(\mathbf{x}_t)), c_1\right) + s_2 \nabla_{\mathbf{x}_t} \ell\left(f_\phi(y|\hat{\mathbf{z}}_0(\mathbf{x}_t)), c_2\right). \quad (3)$$

In Fig. 2, we present generated samples, with goldfish generations in the upper row and tiger sharks in the bottom one (additional samples are provided in Appendix G). By integrating guidance to both $c_1$ and $c_2$ classes, we generate samples from the diffusion model's training data distribution – all of the examples are either goldfishes or sharks but with visible features from classes $c_2$, which are unknown to the diffusion model (e.g., oblong shape of baguette, hay or snow in the background).

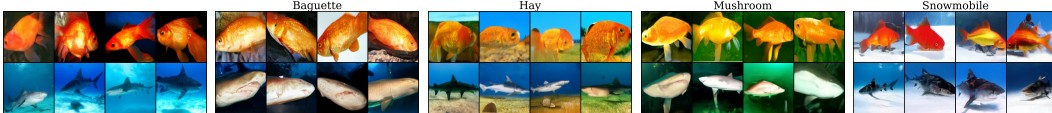

Figure 2: **Samples from the unconditional diffusion model trained only on *goldfish* and *tiger shark* classes from the ImageNet100 dataset.** In the upper row, we present the samples guided to the goldfish class, while in the bottom row, to the tiger shark class. At the same time, the classifier guides the denoising process toward the class depicted above each figure that was not included in the training set of the diffusion model. For reference, in the leftmost column, we present samples generated without guidance toward any unknown class, setting $s_2 = 0$. In every other column, we set both $s_1$ and $s_2$ to 10. We obtain samples from the desired class with observable features of classes unknown to the diffusion model, such as the color, background, or shape.

In this work, we propose to use this observation in the continual training of a classifier, with the distinction that in GUIDE, we only utilize guidance toward unknown classes since we sample from a class-conditional diffusion model. Thus, we eliminate the need for guidance toward classes from the diffusion model's training set.

## 3.2 GUIDANCE TOWARDS CLASSES FROM CURRENT TASK

In this work, we focus on class-incremental continual learning (van de Ven et al., 2022) of a classifier $f_\phi(y|\mathbf{x})$ with generative replay, where we use diffusion model $\epsilon_\theta(\mathbf{x}_t, t, y)$ as a generator of synthetic samples from previous tasks. The data stream comprises of $T$ distinct tasks with disjoint sets of labels. Each task $i$ has its associated dataset $\mathcal{D}_i$ of labeled samples pairs $(\mathbf{x}, y)$.

For each $i \in [1, \ldots, T]$, we train a classifier $f_{\phi_i}(y|\mathbf{x})$ on all tasks up to $i$, using the following loss function:

$$\mathcal{L}_{\leq i}(\mathcal{M}_1, \ldots, \mathcal{M}_{i-1}) = \sum_{k=1}^{i-1} \sum_{(\mathbf{x},y) \in \mathcal{M}_k} \ell(f_{\phi_i}(\mathbf{x}), y) + \sum_{(\mathbf{x},y) \in \mathcal{D}_i} \ell(f_{\phi_i}(\mathbf{x}), y), \quad (4)$$

where $\ell$ is a cross-entropy loss function and $\mathcal{M}_k \subset \mathcal{D}_k$ are free parameters. We will be interested in choosing the sets $\mathcal{M}_k$ to minimize the error in approximating the loss in Equation (4):

$$\mathcal{L}_{\leq i}(\mathcal{D}_1, \ldots, \mathcal{D}_{i-1}) - \mathcal{L}_{\leq i}(\mathcal{M}_1, \ldots, \mathcal{M}_{i-1}) = \sum_{k=1}^{i-1} \sum_{(\mathbf{x},y) \in \mathcal{D}_k \setminus \mathcal{M}_k} \ell(f_{\phi_i}(\mathbf{x}), y). \quad (5)$$

The leading idea of this paper is to choose elements in $(\mathbf{x}, y) \in \mathcal{M}_k$ with large $\ell(f_{\phi_i}(\mathbf{x}), y)$, thus minimizing Equation (5). This demonstrates the power of using a generative replay: we can adaptively correct the *current* version of the classifier, $f_{\phi_i}$, by actively querying for data points that are being misclassified due to continual learning procedure. This would not have been possible if memory-based rehearsal methods were to be used.

More concretely, in the CL scenario, where we only have direct access to data from the current task $\mathcal{D}_i$ but to the previous tasks $\mathcal{D}_1, \ldots, \mathcal{D}_{i-1}$ only through the generative replay $\hat{\epsilon}_{\theta_{i-1}}$, we generate $\mathcal{M}_k$ by sampling from diffusion guided towards examples with a large magnitude of cross-entropy loss:

$$\hat{\epsilon}_{\theta_{i-1}}(\mathbf{x}_t, t, y_{i-1}) = \epsilon_{\theta_{i-1}}(\mathbf{x}_t, t, y_{i-1}) + s\nabla_{\mathbf{x}_t}\ell\left(f_{\phi_i}(y|\hat{\mathbf{z}}_0(\mathbf{x}_t)), y_i\right). \tag{6}$$

Since task $i$ can contain many classes, in each denoising step $t$, we select class $y_i$ from a current task that at that moment yields the highest output from the classifier:

$$y_i = \underset{c \in \mathcal{C}_i}{\operatorname{argmax}} f_{\phi_i}(c|\hat{\mathbf{z}}_0(\mathbf{x}_t)), \tag{7}$$

where $\mathcal{C}_i$ denotes the set of classes in current task $i$.

At the end of each task, we train a class-conditional diffusion model $\epsilon_{\theta_i}(\mathbf{x}_t, t, y)$ on currently available data $\mathcal{D}_i$ along with synthetic data samples from preceding tasks generated by the previous diffusion model $\epsilon_{\theta_{i-1}}(\mathbf{x}_t, t, y)$.

Intuitively, in GUIDE, we steer the diffusion process towards examples from the current task, as depicted in Fig. 1. Simultaneously, since we utilize only the previous frozen diffusion model that is not trained on classes from the current task, we consistently obtain samples from desired class $y_{i-1}$. Rehearsal examples obtained with the modified sampling process yield lower outputs for class $y_{i-1}$ in the current classifier $f_{\phi_i}(y|\mathbf{x})$ compared to the previous classifier $f_{\phi_{i-1}}(y|\mathbf{x})$. Hence, these examples can be interpreted as data samples that are more likely to be forgotten during continual training. We experimentally validate this statement in Sec. 4.5.

The effect of our guidance technique resembles the idea introduced in Retrospective Adversarial Replay (RAR) (Kumari et al., 2022), where authors show that training the classifier on rehearsal samples similar to examples from the current task helps the model to learn the boundaries between tasks. This is also in line with observation by Toneva et al. (2018), who show that the sample's distance from the decision border is correlated with the number of forgetting events.

## 4 EXPERIMENTS

### 4.1 EXPERIMENTAL SETUP

**Datasets** We evaluate our approach on CIFAR-10 and CIFAR-100 (Krizhevsky, 2009) image datasets. We split the CIFAR-10 dataset into 2 and 5 equal tasks and the CIFAR-100 dataset into 5 and 10 equal tasks. Moreover, to validate if our method can be extended to datasets with higher resolution, we also evaluate it on the ImageNet-100 (Deng et al., 2009) dataset split into 5 tasks.

**Metrics** For evaluation on each task $i \in [1, \ldots, T]$, we use two metrics commonly used in continual learning: average accuracy $\bar{A}_i = \frac{1}{i}\sum_{j=1}^{i} A_j^i$ and average forgetting $\bar{F}_i = \frac{1}{i-1}\sum_{j=1}^{i-1} \max_{1 \le k \le i}(A_j^k - A_j^i)$, where $A_j^i$ denotes the accuracy of a model on $j$-th task after training on $i$ tasks. We follow the definitions of both metrics from (Chaudhry et al., 2018).

**Baseline methods** We compare our approach with state-of-the-art generative replay methods. For fairness, in the evaluation of **BIR** (Van de Ven et al., 2020), we freeze the encoder model after training on the first task. In order to evaluate **MeRGAN** (Wu et al., 2018) on generative replay scenario, we generate samples from the final generator network trained sequentially on all tasks to construct a training dataset for the ResNet18 classifier. We also compare the results to the **Joint** training on all data and simple **Fine-tuning** with no rehearsal. As a soft upper bound of the proposed method, we present a **Continual Joint** setting, where we train the classifier continually with full access to all previous tasks (perfect rehearsal with infinite buffer size). In all methods using the diffusion model (including **DDGR**), we use the same number of denoising steps to obtain rehearsal samples. We recalculated scores for all related methods using the code provided by the authors. Importantly, we do not use any pre-training on external datasets.

### 4.2 IMPLEMENTATION DETAILS

Our training procedure is divided into two parts. First, in each task $i$, we train the classifier $f_{\phi_i}(y|\mathbf{x})$. In order to generate rehearsal samples, we load the diffusion model $\epsilon_{\theta_{i-1}}(\mathbf{x}_t, t, y)$ already trained on the previous task. Then, we train a class-conditional diffusion model $\epsilon_{\theta_i}(\mathbf{x}_t, t, y)$ in a standard

Table 1: Comparison of GUIDE with other generative rehearsal methods (we mark feature replay methods in gray color). Our approach outperforms most other methods in terms of both average accuracy and average forgetting after the final task $T$.

| | AVERAGE ACCURACY $\bar{A}_T$ (↑) | | | | | AVERAGE FORGETTING $\bar{F}_T$ (↓) | | | | |
|---|---|---|---|---|---|---|---|---|---|---|
| | CIFAR-10 | | CIFAR-100 | | IMAGENET100 | CIFAR-10 | | CIFAR-100 | | IMAGENET100 |
| METHOD | $T=2$ | $T=5$ | $T=5$ | $T=10$ | $T=5$ | $T=2$ | $T=5$ | $T=5$ | $T=10$ | $T=5$ |
| JOINT | 93.14 ± 0.16 | | 72.32 ± 0.24 | | 66.85 ± 2.25 | - | - | - | - | - |
| CONTINUAL JOINT | 85.63 ± 0.39 | 86.41 ± 0.32 | 73.07 ± 0.01 | 64.15 ± 0.98 | 50.59 ± 0.35 | 7.91 ± 0.67 | 2.90 ± 0.08 | 7.80 ± 0.55 | 6.67 ± 0.36 | 12.28 ± 0.07 |
| FINE-TUNING | 47.22 ± 0.06 | 18.95 ± 0.20 | 16.92 ± 0.03 | 9.12 ± 0.04 | 13.49 ± 0.18 | 92.69 ± 0.06 | 94.65 ± 0.17 | 80.75 ± 0.22 | 87.67 ± 0.07 | 64.93 ± 0.00 |
| DGR VAE | 60.24 ± 1.53 | 28.23 ± 3.84 | 19.66 ± 0.27 | 10.04 ± 0.17 | 9.54 ± 0.26 | 43.91 ± 5.40 | 57.21 ± 9.82 | 42.10 ± 1.40 | 60.31 ± 4.80 | 40.26 ± 0.91 |
| DGR+DISTILL | 52.40 ± 2.58 | 27.83 ± 1.20 | 21.38 ± 0.61 | 13.94 ± 0.13 | 11.77 ± 0.47 | 70.84 ± 6.35 | 43.43 ± 2.60 | 29.30 ± 0.40 | 21.15 ± 1.30 | 41.17 ± 0.43 |
| RTF | 51.80 ± 2.56 | 30.36 ± 1.40 | 17.45 ± 0.28 | 12.80 ± 0.78 | 8.03 ± 0.05 | 60.49 ± 5.54 | 51.77 ± 1.00 | 47.68 ± 0.80 | 45.21 ± 5.80 | 41.2 ± 0.20 |
| MeRGAN | 50.54 ± 0.08 | 51.65 ± 0.40 | 9.65 ± 0.14 | 12.34 ± 0.15 | - | - | - | - | - | - |
| BIR | 53.97 ± 0.97 | 36.41 ± 0.82 | 21.75 ± 0.08 | 15.26 ± 0.49 | 8.63 ± 0.19 | 64.97 ± 2.15 | 65.28 ± 1.27 | 48.38 ± 0.44 | 53.08 ± 0.75 | 40.99 ± 0.36 |
| GFR | 64.13 ± 0.88 | 26.70 ± 1.90 | 34.80 ± 0.26 | 21.90 ± 0.14 | 32.95 ± 0.35 | 25.37 ± 6.62 | 49.29 ± 6.03 | 19.16 ± 0.55 | 17.44 ± 2.20 | 20.37 ± 1.47 |
| DDGR | 80.03 ± 0.65 | 43.69 ± 2.60 | 28.11 ± 2.58 | 15.99 ± 1.08 | 25.59 ± 2.29 | 22.45 ± 1.13 | 62.51 ± 3.84 | 60.62 ± 2.13 | 74.70 ± 1.79 | 49.52 ± 2.52 |
| DGR DIFFUSION | 77.43 ± 0.60 | 59.00 ± 0.57 | 28.25 ± 0.22 | 15.90 ± 1.01 | 23.92 ± 0.92 | 26.32 ± 0.90 | 40.38 ± 0.32 | 68.70 ± 0.65 | 80.38 ± 1.34 | 54.44 ± 0.14 |
| **GUIDE** | **81.29 ± 0.75** | **64.47 ± 0.45** | **41.66 ± 0.40** | **26.13 ± 0.29** | **39.07 ± 1.37** | **14.79 ± 0.36** | **24.84 ± 0.05** | 44.30 ± 1.10 | 60.54 ± 0.82 | 27.60 ± 3.28 |

self-rehearsal approach, independently of the classifier. In the first task, we train both the diffusion model and the classifier solely on real data samples.

In our method, it is essential to generate rehearsal examples progressively as the classifier is trained on the task. This way, similarly to active learning techniques, at each step, we guide the generation process toward the most challenging samples – those that are close to the decision boundary of the classifier at the given moment. The mini-batches are balanced to ensure that each contains an equal number of samples from each class encountered so far. For clarity, we present a detailed pseudocode of the entire training procedure in Appendix C. All training hyperparameters can be found in Appendix D and in the code repository[1].

## 4.3 EXPERIMENTAL RESULTS

We evaluate our method on a set of CL benchmarks with a comparative analysis conducted in relation to other generative rehearsal techniques. In Tab. 1, we present the mean and standard deviation of results calculated for 3 random seeds. Our method outperforms other evaluated methods regarding the average accuracy after the last task $\bar{A}_T$ by a considerable margin on all benchmarks. Specifically, GUIDE notably improves upon the standard DGR with diffusion model on both average incremental accuracy and forgetting. It also outperforms DDGR, another GR approach that uses the diffusion model, proving the superiority of our sampling technique.

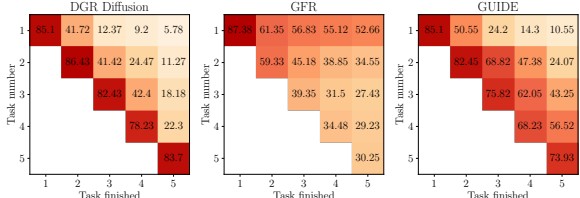

Figure 3: **Accuracy on each task during each phase of class-incremental training on CIFAR-100 with 5 tasks - standard GR with diffusion (left), GFR (middle), and our method (right).** We observe the stability-plasticity trade-off, where our method significantly reduces forgetting compared to the standard GR scenario at the cost of a slight decrease in the ability to learn new tasks.

We further compare our method to feature replay methods (BIR and GFR). In the case of the GFR, we observe less forgetting on the CIFAR-100 and ImageNet100 datasets. This is the effect of the drastically limited plasticity of GFR due to the decrease in the number of updates to the feature extractor after the first task. For a more detailed analysis, in Fig. 3 we thoroughly benchmark our method against the standard GR scenario and GFR method, presenting the accuracy on each encountered task after each training phase on CIFAR-100 with 5 tasks. Our approach significantly improved upon the standard GR scenario in terms of knowledge retention from preceding tasks. It indicates that training on rehearsal examples generated by GUIDE successfully mitigates forgetting. The decreased accuracy on the most recent task in our method can be interpreted through the lens of the stability-plasticity trade-off (Grossberg, 1982), highlighting that while our approach substantially reduces the forgetting of classifier, it does so at the expense of its ability to assimilate new information. Although GFR can maintain the performance on the initial task notably better than our method thanks to limited training of feature extractor on subsequent tasks, it leads to a substantial final performance drop.

---

[1]Link hidden for review

Table 2: Evaluation of GUIDE in the online scenario. The memory buffer size for each ER method is set to match the size of the diffusion model used in the GUIDE method. In the DGR diffusion and GUIDE, we use a diffusion model trained on all data samples from previous tasks.

| | | AVERAGE ACCURACY $\bar{A}_T$ ($\uparrow$) | | AVERAGE FORGETTING $\bar{F}_T$ ($\downarrow$) | |
|---|---|---|---|---|---|
| **METHOD** | **BUFFER SIZE** $m$ | CIFAR-10 $T = 5$ | CIFAR-100 $T = 5$ | CIFAR-10 $T = 5$ | CIFAR-100 $T = 5$ |
| JOINT | - | $50.34 \pm 0.78$ | $18.67 \pm 0.52$ | - | - |
| CONTINUAL JOINT | $\infty$ | $36.94 \pm 0.48$ | $20.57 \pm 0.47$ | $37.92 \pm 1.12$ | $17.25 \pm 0.71$ |
| FINE-TUNING | 0 | $13.68 \pm 0.40$ | $2.82 \pm 0.59$ | $71.78 \pm 2.75$ | $28.53 \pm 0.68$ |
| ER | 16 573 | $36.56 \pm 2.18$ | $13.45 \pm 1.09$ | $41.77 \pm 2.64$ | $27.77 \pm 2.54$ |
| iCaRL | 16 573 | $33.48 \pm 4.20$ | $8.62 \pm 0.16$ | $\underline{31.87} \pm 3.22$ | $\mathbf{8.97} \pm 0.38$ |
| GSS | 16 573 | $18.70 \pm 0.23$ | $15.04 \pm 3.23$ | $83.78 \pm 0.62$ | $41.75 \pm 4.06$ |
| FDR | 16 573 | $18.46 \pm 0.09$ | $11.13 \pm 0.28$ | $82.08 \pm 0.64$ | $46.05 \pm 0.67$ |
| DER | 16 573 | $26.45 \pm 3.82$ | $8.09 \pm 0.74$ | $49.41 \pm 5.03$ | $40.17 \pm 1.08$ |
| DER++ | 16 573 | $\underline{40.73} \pm 2.64$ | $\mathbf{25.94} \pm 1.01$ | $\mathbf{24.77} \pm 3.89$ | $22.64 \pm 1.32$ |
| DGR DIFFUSION* | - | $38.83 \pm 2.50$ | $21.55 \pm 0.41$ | $42.95 \pm 6.38$ | $16.72 \pm 1.13$ |
| **GUIDE*** | - | $\mathbf{43.30} \pm 0.57$ | $\underline{22.78} \pm 0.94$ | $35.49 \pm 0.57$ | $\underline{11.75} \pm 2.93$ |

## 4.4 EVALUATION IN THE ONLINE SCENARIO

In GUIDE, we train the diffusion model, which requires multiple passes through the dataset to achieve satisfactory performance, naturally classifying it as an offline CL method. However, to fairly evaluate our sampling strategy against buffer-based Experience Replay (ER) approaches, we conduct experiments where the diffusion model is trained offline on all the data from previous tasks, while the classifier is trained in an online scenario, where each data sample is seen only once during the training.

To ensure a fair comparison, the buffer size for each ER method is matched to the size of the diffusion model used in our method on the CIFAR datasets. Detailed hyperparameters for each ER baseline method are provided in Appendix B.

As baseline buffer-based methods, we evaluate **ER** (Riemer et al., 2019), **iCaRL** (Rebuffi et al., 2017), **GSS** (Aljundi et al., 2019b), **FDR** (Benjamin et al., 2019), **DER** and **DER++** (Buzzega et al., 2020). As shown in Tab. 2, GUIDE, despite not being initially designed for the online scenario, performs on par with the evaluated ER approaches. Notably, our sampling approach outperforms the **Continual Joint** setup, which served as a soft upper bound in our offline evaluation. This demonstrates that in scenarios with a very limited number of model updates, rehearsal samples produced by GUIDE are more informative for CL purposes than real data samples stored in an infinite buffer.

## 4.5 ANALYSIS OF THE PROPOSED GUIDANCE

To demonstrate that our method produces samples near the decision boundary of a classifier, we propose to launch a simple adversarial attack on the generated samples in order to check how easy it is to change their class to the one from the current task. Concretely, we adapt the method introduced by Goodfellow et al. (2015), and modify each rehearsal sample that was generated during training $\hat{\mathbf{x}}$ as follows:

Table 3: Proportion of misclassified rehearsal samples after the perturbation. Samples generated via GUIDE exhibit a higher misclassification rate, signifying their proximity to the classifier's decision boundary. Moreover, rehearsal samples in our method yield lower outputs for both previous and current classifiers.

$$\hat{\mathbf{x}}^* = \hat{\mathbf{x}} - \epsilon \text{sign} \left( \nabla_{\hat{\mathbf{x}}} \ell(f_{\phi_i}(y|\hat{\mathbf{x}}), y_i) \right), \qquad (8)$$

where $\epsilon = 0.1$ and $y_i = \underset{c \in \mathcal{C}_i}{\arg\max} f_{\phi_i}(y = c|\hat{\mathbf{x}})$.

| | MISCLASSIFIED EXAMPLES | CONFIDENCE PREV | CURR |
|---|---|---|---|
| DGR DIFFUSION | 55.13% | 99.6% | 90.03% |
| GUIDE | 72.66% | 86.42% | 61.61% |

Then, we calculate the proportion of cases where the classifier's prediction for the modified sample $\hat{\mathbf{x}}^*$ differs from its prediction for the original generated sample $\hat{\mathbf{x}}$. As shown in Tab. 3, we can change the classifier's prediction much more frequently when we sample the replay examples according to our method. Since rehearsal examples generated with GUIDE are much more likely to be misclassified after a simple modification with predefined magnitude, this indicates that the modification of diffusion's prediction introduced in our method successfully moves the generations closer to the classifier's decision boundary.

Table 4: Comparison of evaluated variants of integrating classifier guidance in CL. **PREV** and **CURR** refers to guidance from previous and current classifier respectively. Guidance toward classes from the previous tasks is denoted with "-" and guidance towards classes from the current task with "+". Each of the introduced variants outperforms standard DGR with diffusion model on most of the evaluated benchmarks and achieves state-of-the-art performance.

| | AVERAGE ACCURACY $\bar{A}_T$ ($\uparrow$) | | | | | AVERAGE FORGETTING $\bar{F}_T$ ($\downarrow$) | | | | |
| | CIFAR-10 | | CIFAR-100 | | IMAGENET100 | CIFAR-10 | | CIFAR-100 | | IMAGENET100 |
| VARIANT | $T=2$ | $T=5$ | $T=5$ | $T=10$ | $T=5$ | $T=2$ | $T=5$ | $T=5$ | $T=10$ | $T=5$ |
|---|---|---|---|---|---|---|---|---|---|---|
| DGR DIFFUSION | $77.43 \pm 0.60$ | $59.00 \pm 0.57$ | $28.25 \pm 0.22$ | $15.90 \pm 1.01$ | $23.92 \pm 0.92$ | $26.32 \pm 0.90$ | $40.38 \pm 0.32$ | $68.70 \pm 0.65$ | $80.38 \pm 1.34$ | $54.44 \pm 0.14$ |
| PREV + | $80.03 \pm 0.65$ | $60.31 \pm 0.44$ | $31.35 \pm 0.66$ | $18.22 \pm 0.52$ | $29.13 \pm 2.81$ | $22.45 \pm 1.13$ | $40.00 \pm 0.60$ | $64.80 \pm 1.00$ | $77.60 \pm 0.50$ | $45.15 \pm 3.64$ |
| PREV - | $75.79 \pm 1.20$ | $57.05 \pm 0.43$ | $28.40 \pm 0.04$ | $15.79 \pm 0.21$ | $12.60 \pm 0.71$ | $27.58 \pm 0.95$ | $44.80 \pm 0.90$ | $68.34 \pm 0.29$ | $80.42 \pm 0.44$ | $59.98 \pm 0.28$ |
| CURR - | $78.72 \pm 0.58$ | $57.72 \pm 0.95$ | $30.57 \pm 0.33$ | $16.87 \pm 0.83$ | $23.55 \pm 3.27$ | $23.89 \pm 0.44$ | $43.31 \pm 1.50$ | $65.42 \pm 0.81$ | $78.96 \pm 1.16$ | $53.85 \pm 3.60$ |
| **GUIDE** | $\mathbf{81.29} \pm 0.75$ | $\mathbf{64.47} \pm 0.45$ | $\mathbf{41.66} \pm 0.40$ | $\mathbf{26.13} \pm 0.29$ | $\mathbf{39.07} \pm 1.37$ | $\mathbf{14.79} \pm 0.36$ | $\mathbf{24.84} \pm 0.05$ | $\mathbf{44.30} \pm 1.10$ | $\mathbf{60.54} \pm 0.82$ | $\mathbf{27.60} \pm 3.28$ |

Moreover, we present a visualization of generated samples in the latent space of a classifier (Fig. 4) that we calculate during the training of the second task on a CIFAR-10 dataset divided into five equal tasks. In the standard GR scenario, the rehearsal samples originate predominantly from high-density regions of class manifolds, which is evident from their central location within each class's manifold. On the other hand, our method yields generations that are more similar to the examples from the second task.

## 5 ADDITIONAL ANALYSIS

### 5.1 THE EFFECT OF CHANGING CLASSIFIER SCALE

An important hyperparameter of our method is the gradient scaling parameter $s$ that controls the strength of the guidance signal. In this section, we show the effect of the gradient scale $s$ on the effectiveness of our method. However, our method is robust and can work well with different values of this parameter. The detailed results with other scaling parameters are presented in Appendix A. As we increase the scale, we observe the inflection point, after which the accuracy starts to drop. This is related to the observation that with excessively large scaling parameters, the quality of generated samples drops significantly.

In Fig. 5, we present results averaged over 3 random seeds for different values of gradient scale $s$, presenting standard deviation as error bars. We observe that the scaling parameter introduces a trade-off between the stability and plasticity of the continually trained classifier. When we increase $s$, the accuracy on the previous task increases along with a drop in accuracy on the current task.

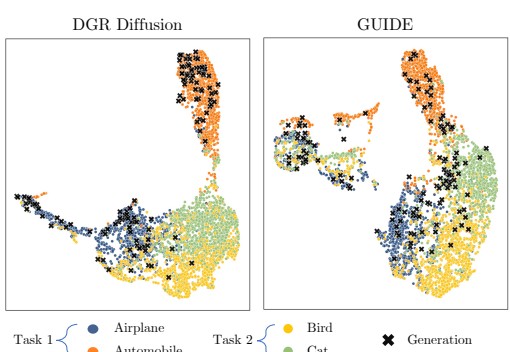

Figure 4: **Visualization of the classifiers embedding space (umap) for training examples and generations sampled with standard generative replay method (left) and ours (right) at 75% of the training on the second task.** We can observe how GUIDE sample generations are more similar to the training examples from new classes (e.g., airplanes similar to birds).

### 5.2 ALTERNATIVE VARIANTS OF GUIDANCE

In addition to our primary method, we evaluate alternative variants of incorporating classifier guidance to generative replay setup, drawing inspiration from corresponding techniques in buffer-based rehearsal. Each variant effectively modifies the sampling strategy from the diffusion model. In each variant, we benefit either from the frozen classifier $f_{\phi_{i-1}}(y|\mathbf{x})$, trained on prior tasks and henceforth referred to as the *previous classifier*, or the currently trained classifier, $f_{\phi_i}(y|\mathbf{x})$. In this section, we define each variant highlighting with the blue color the distinctions from GUIDE.

**Guidance towards classes from previous tasks** The most straightforward adaptation of a classifier guidance concept to a generative replay setup is to modify the diffusion sampling process using

the gradients from the previous frozen classifier to refine the quality of rehearsal samples. The modification of the previous diffusion model's prediction can be thus defined as:

$$\hat{\epsilon}_{\theta_{i-1}}(\mathbf{x}_t, t, y_{i-1}) = \epsilon_{\theta_{i-1}}(\mathbf{x}_t, t, y_{i-1}) + s\nabla_{\mathbf{x}_t}\ell\left(f_{\phi_{i-1}}(y|\hat{\mathbf{z}}_0(\mathbf{x}_t)), y_{i-1}\right),\qquad(9)$$

where $y_{i-1}$ denotes the class label from one of the previous tasks. As noted by Dhariwal & Nichol (2021), the application of classifier guidance creates a trade-off: it enhances the quality of the generated samples at the cost of their diversity. This approach is similar to the one introduced by Gao & Liu (2023) except that we do not use guidance in the process of continual diffusion training but only in the classifier's training. Intuitively similar buffer-based methods are based on the herding algorithm (Welling, 2009) and used in iCaRL method (Rebuffi et al., 2017), which seeks to store samples that best represent the mean of classes in the feature space.

**Guidance away from classes from previous tasks**   Alternatively, we can guide the diffusion-denoising process in the opposite direction by maximizing the entropy of the classifier instead of minimizing it. As noted by Sehwag et al. (2022), such an approach steers the denoising diffusion process away from the high-density regions of the data manifold. Consequently, it should generate synthetic samples that resemble the rare instances in the training dataset, which are typically more challenging for the classifier to identify. In the first variant, we propose to use the previous classifier $f_{\phi_{i-1}}(y|\mathbf{x})$ to guide away from the old classes:

$$\hat{\epsilon}_{\theta_{i-1}}(\mathbf{x}_t, t, y_{i-1}) = \epsilon_{\theta_{i-1}}(\mathbf{x}_t, t, y_{i-1}) - s\nabla_{\mathbf{x}_t}\ell\left(f_{\phi_{i-1}}(y|\hat{\mathbf{z}}_0(\mathbf{x}_t)), y_{i-1}\right).\qquad(10)$$

Analogously, we can steer the diffusion denoising process away from the desired class from the previous task, but using the current classifier $f_{\phi_i}(y|\mathbf{x})$:

$$\hat{\epsilon}_{\theta_{i-1}}(\mathbf{x}_t, t, y_{i-1}) = \epsilon_{\theta_{i-1}}(\mathbf{x}_t, t, y_{i-1}) - s\nabla_{\mathbf{x}_t}\ell\left(f_{\phi_i}(y|\hat{\mathbf{z}}_0(\mathbf{x}_t)), y_{i-1}\right).\qquad(11)$$

In both approaches, we increase the diversity of rehearsal samples under frozen or continually trained classifiers. This variation resembles the buffer-based method of Gradient Sample Selection (GSS) (Aljundi et al., 2019b), which seeks to maximize the diversity of the samples stored in the memory buffer.

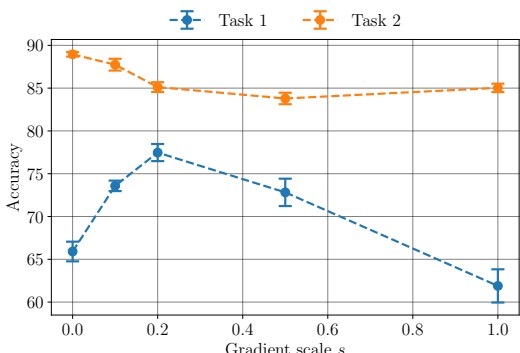

**Evaluation**   We evaluate the performance of each proposed variant and present the mean and standard deviation of those experiments on CL benchmarks in Tab. 4, which are calculated for 3 random seeds. We can observe that thanks to the improved quality of rehearsal samples, approaches integrating guidance towards selected classes achieve higher overall performance. In simpler scenarios, increasing the diversity of generated samples can also yield slight improvement, while in more complex settings, its performance is comparable to the baseline. Nevertheless, in all evaluated scenarios, our proposed guidance towards forgotten examples outperforms competing approaches.

Figure 5: **Classifier scale impact on forgetting and ability to acquire new information.** Up to $s = 0.2$, when we increase scale, we reduce the forgetting but also observe a drop in the accuracy on the second task. When we use too large scale $s$, the quality of samples drops significantly, along with the accuracy on the previous task. We further present this effect in Appendix A.

### 5.3   IMPACT OF FORGETTING IN THE DIFFUSION MODEL ON GUIDE

The aim of this work is to sample from a diffusion model in the most informative way for the continual training of a classifier. However, to continually train the diffusion model itself, we employ a simple self-rehearsal approach. To examine the effect of forgetting in the generative model on the performance of GUIDE, we independently train four diffusion models on all data presented up to the $i^{th}$ task in the CIFAR-100/5 scenario. Using these models, we compare our method with the baseline sampling method and the continual upper bound.

The results of this comparison are shown in Fig. 6, calculated across 3 random seeds with the mean and standard deviation depicted as error bars. Our fully continual learning setup (*GUIDE*) still outperforms the baseline sampling approach, which uses a diffusion model trained on all previous data samples (*DGR Diffusion\**), indicating that the random sampling method causes a significant drop in performance.

Importantly, there is a significant difference between the *GUIDE* and *GUIDE\** setups, indicating that forgetting in the continually trained diffusion model is the main factor of the classifier's performance loss. We will investigate this aspect of forgetting in future works.

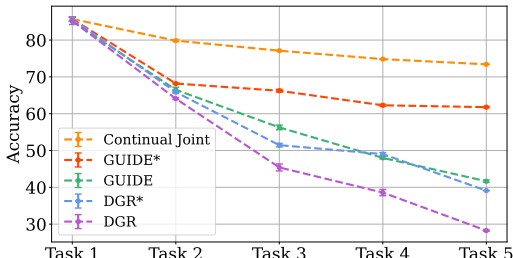

Figure 6: **Average accuracy on all classes seen so far after $i^{th}$ task.** We highlight with asterisks $(*)$ methods trained using diffusion model trained on all previous data samples. We observe that when eliminating the forgetting in the diffusion model, GUIDE approaches the soft upper-bound defined by the continual joint training.

## 5.4 COMPUTATIONAL COST OF GUIDE

A significant drawback of all generative replay approaches is the computational burden associated with training and sampling from the diffusion model. Therefore, we explore two simple speedup techniques that allow us to reduce computation costs notably. While all diffusion models in this work are trained with 1000 steps, we can change the sampler to DDIM (Song et al., 2020) with fewer sampling steps. To measure the effect of this approach, in Tab. 5, we evaluate how limiting the number of backward diffusion steps affects the final average accuracy on the CIFAR10/5 benchmark. We can observe an order of magnitude speedup at the expense of a slight drop in the model's performance. Therefore, in all of the experiments presented in this work, we limit the number of sampling steps to 50 or 100, as mentioned in Appendix D. Similarly, in Tab. 6, we measure the interval (in numbers of batches) between which we sample new generations. We observe that sampling rehearsal examples once every five batches yields similar results to the default version while providing a 5-times speedup. Combining those two simple techniques allows us to achieve an 83 times faster generation process. We provide a detailed comparison of training times in Appendix F, showing that our approach is more computationally expensive than VAE and GAN-based techniques but, at the same time, over 25% faster than the existing state-of-the-art diffusion-based method (DDGR).

Table 5: Trade-off between the number of denoising steps used for the rehearsal sampling on the effectiveness of GUIDE on CIFAR-10/5.

| $N^o$ steps | $\bar{A}_T$ | Time (speedup) |
|---|---|---|
| 10 | $59.7 \pm 1.04$ | $\times 7.3$ |
| 20 | $63.7 \pm 0.99$ | $\times 2.3$ |
| 50 | $64.1 \pm 0.09$ | $\times 1$ |
| 100 | $64.2 \pm 0.54$ | $\times 0.51$ |
| 250 | $63.6 \pm 0.25$ | $\times 0.21$ |
| 1000 | $71.8 \pm 0.23$ | $\times 0.06$ |

Table 6: Effect of the interval between new rehearsal samples generation on the effectiveness of GUIDE on CIFAR-10/5.

| Generation interval | $\bar{A}_T$ | Time (speedup) |
|---|---|---|
| 1 | $63.1 \pm 0.45$ | $\times 0.2$ |
| 5 | $64.5 \pm 0.45$ | $\times 1$ |
| 10 | $64.1 \pm 0.09$ | $\times 1.95$ |
| 50 | $59.2 \pm 0.47$ | $\times 8.49$ |
| 100 | $56.9 \pm 0.98$ | $\times 14.41$ |
| $\infty$ | $18.9 \pm 0.20$ | $\times 49.59$ |

## 6 CONCLUSION

In this work, we propose GUIDE: generative replay method that utilizes classifier guidance to generate rehearsal samples that the classifier model is likely to forget. We benefit from a classifier trained continually in each task to guide the denoising diffusion process toward the most recently encountered classes. This strategy enables the classifier's training with examples near its decision boundary, rendering them particularly valuable for continual learning. Across various CL benchmarks, GUIDE demonstrates superior performance, consistently surpassing recent state-of-the-art generative rehearsal methods. This underscores the effectiveness of our approach in mitigating forgetting and training a robust classifier.

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

## A  IMPACT OF CLASSIFIER SCALE HYPERPARAMETER ON OUR METHOD

To measure the effect of changing the strength of guidance scale parameter $s$ on the effectiveness of our method, we sweep over each value in $[0.1, 0.2, 0.5, 1.0]$ for each evaluated benchmark. We present the mean and standard deviation of results in Tab. 7 calculated for 3 different values of random seed. We see that with this hyperparameter properly tuned, we are able to improve the results of our method further. Nonetheless, the results we achieved show that our method works well with different values of this hyperparameter.

Table 7: Effect of gradient scale $s$ on GUIDE. By tuning this hyperparameter, we can improve the results further.

| | AVERAGE ACCURACY $\bar{A}_T$ ($\uparrow$) | | | | | AVERAGE FORGETTING $\bar{F}_T$ ($\downarrow$) | | | | |
| | CIFAR-10 | | CIFAR-100 | | IMAGENET100-64 | CIFAR-10 | | CIFAR-100 | | IMAGENET100-64 |
| SCALE $s$ | $T = 2$ | $T = 5$ | $T = 5$ | $T = 10$ | $T = 5$ | $T = 2$ | $T = 5$ | $T = 5$ | $T = 10$ | $T = 5$ |
|---|---|---|---|---|---|---|---|---|---|---|
| 0.0 | $77.43 \pm 0.60$ | $56.61 \pm 1.85$ | $28.25 \pm 0.22$ | $15.90 \pm 1.00$ | $23.92 \pm 0.92$ | $26.32 \pm 0.90$ | $43.79 \pm 0.41$ | $68.70 \pm 0.65$ | $80.38 \pm 1.34$ | $54.44 \pm 0.14$ |
| 0.1 | $80.66 \pm 0.44$ | $59.56 \pm 0.52$ | $31.63 \pm 0.81$ | $18.28 \pm 0.93$ | $26.48 \pm 3.79$ | $18.81 \pm 0.48$ | $39.78 \pm 0.92$ | $63.87 \pm 1.35$ | $78.01 \pm 0.59$ | $50.09 \pm 5.04$ |
| 0.2 | $\mathbf{81.29 \pm 0.75}$ | $61.30 \pm 0.10$ | $37.51 \pm 1.23$ | $22.68 \pm 0.30$ | $31.09 \pm 4.17$ | $14.79 \pm 0.36$ | $34.98 \pm 0.13$ | $54.52 \pm 1.45$ | $70.68 \pm 0.23$ | $44.40 \pm 5.02$ |
| 0.5 | $78.30 \pm 0.47$ | $\mathbf{64.47 \pm 0.45}$ | $\mathbf{41.66 \pm 0.40}$ | $25.48 \pm 1.16$ | $35.82 \pm 0.56$ | $20.03 \pm 1.85$ | $29.25 \pm 1.15$ | $44.30 \pm 1.10$ | $64.06 \pm 1.53$ | $35.80 \pm 0.32$ |
| 1.0 | $73.46 \pm 0.81$ | $62.56 \pm 0.52$ | $39.55 \pm 0.19$ | $\mathbf{26.13 \pm 0.29}$ | $\mathbf{39.07 \pm 1.37}$ | $30.96 \pm 1.93$ | $\mathbf{24.05 \pm 0.72}$ | $\mathbf{44.03 \pm 0.34}$ | $\mathbf{60.54 \pm 0.82}$ | $\mathbf{27.60 \pm 3.29}$ |

Moreover, in Fig. 7, we present random rehearsal samples from our CIFAR-10 setup with 2 tasks generated from the same initial noise in each column. We observe the effect of classifier scale parameter $s$ on the quality of rehearsal samples in GUIDE. If we set the scale to be too large, we observe significant degradation in the quality of generations. Hence, in Tab. 7, we observe a drop in performance on setups where the scale is too large.

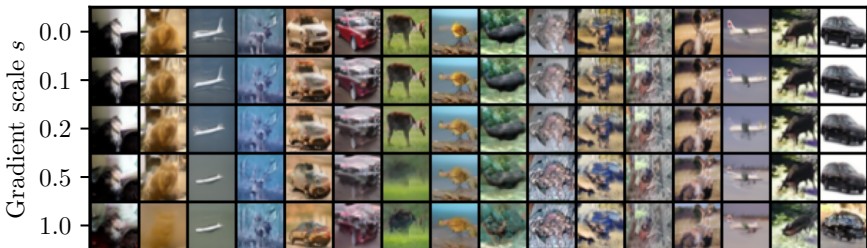

Figure 7: **Sample rehearsal examples of GUIDE generated by class-conditional diffusion model trained on the first task of CIFAR-10/2 setup.** If we set the gradient scale parameter $s$ too large, we observe a significant drop in the quality of samples.

## B  EVALUATION OF ER METHODS

For each ER approach that we evaluate we choose the set of the best hyperparameters reported in Buzzega et al. (2020) for the CIFAR-10 dataset with a buffer size equal to 5120:

- **GSS** (Aljundi et al., 2019b): learning rate: 0.03
- **iCaRL** (Rebuffi et al., 2017): learning rate: 0.03
- **FDR** (Benjamin et al., 2019): learning rate: 0.03, $\alpha$: 0.3
- **DER** (Buzzega et al., 2020): learning rate: 0.03, $\alpha$: 0.3
- **DER++** (Buzzega et al., 2020): learning rate: 0.03, $\alpha$: 0.1, $\beta$: 1.0

We run the training of each approach for 1 epoch and report the results averaged over 3 random seeds. We do not use any data augmentations and normalize the images to the range [-1, 1]. In each method, we use a batch size equal to 32, where half of the batch comes from the current task and half comes from the buffer. We reproduce the results from the GitHub code repository: `https://github.com/aimagelab/mammoth`.

## C  Pseudocode of training procedure in GUIDE

To clarify and enhance understanding of our continual training process, this section includes pseudocode for the primary components involved in our method. Alg. 1 presents a sampling of a single rehearsal example during the continual training of a classifier in GUIDE. In Alg. 2, we present a complete end-to-end continual training of both diffusion model and classifier in GUIDE.

Since the decision boundary changes during the training of a classifier, we sample new rehearsal examples according to our sampling method progressively once every $N_g$ batches. It ensures that replayed examples are located close to the decision boundary of a classifier at the given moment. Simultaneously, hyperparameter $N_g$ allows us to significantly speed up our method without a noticeable drop in accuracy. Since we do not use classifier guidance during the generation of a dataset for diffusion training, we sample rehearsal examples all at once and merge them with real data samples.

In the pseudocode, consistent with the terminology used in the main text, the current classifier is denoted as $f_{\phi_i}(y|\mathbf{x})$ and the previous diffusion model as $\epsilon_{\theta_{i-1}}(\mathbf{x}_t, t, y)$. All hyperparameters, such as the number of training steps or the number of denoising steps, are listed in Appendix D.

---

**Algorithm 1** Rehearsal sampling in GUIDE during training on task $i$

---

**Input:** $\mathcal{C}_{i-1}$: classes from all previous tasks, $\mathcal{C}_i$: classes from current task, $s$: gradient scale, $T$: number of denoising steps

    $t \leftarrow T$
    $y_{i-1} \sim \mathcal{U}(\mathcal{C}_{i-1})$
    $\mathbf{x}_t \sim \mathcal{N}(0, \mathbf{I})$
    **while** $t > 0$ **do**
        $\hat{\mathbf{z}}_0(\mathbf{x}_t) \leftarrow \frac{\mathbf{x}_t - \sqrt{1-\bar{\alpha}_t}\epsilon_{\theta_{i-1}}(\mathbf{x}_t, t, y_{i-1})}{\sqrt{\bar{\alpha}_t}}$
        $y_i \leftarrow \underset{c \in \mathcal{C}_i}{\arg\max} f_{\phi_i}(y = c | \hat{\mathbf{z}}_0(\mathbf{x}_t))$
        $\hat{\epsilon}_{\theta_{i-1}}(\mathbf{x}_t, t, y_{i-1}) \leftarrow \epsilon_{\theta_{i-1}}(\mathbf{x}_t, t, y_{i-1}) + s\nabla_{\mathbf{x}_t}\ell\left(f_{\phi_i}(y|\hat{\mathbf{z}}_0(\mathbf{x}_t), y_i)\right)$
        $\mathbf{x}_{t-1} \leftarrow \sqrt{\bar{\alpha}_{t-1}}\hat{\mathbf{z}}_0(\mathbf{x}_t) + \sqrt{1 - \bar{\alpha}_{t-1}}\hat{\epsilon}_{\theta_{i-1}}(\mathbf{x}_t, t, y_{i-1})$
        $t \leftarrow t - 1$
    **end while**
**Output:** $\mathbf{x}_0$                      ▷ Generated rehearsal sample from class $y_{i-1}$

---

---

**Algorithm 2** Continual training in GUIDE

---

**Input:** $N_c$: number of classifier training steps, $N_d$: number of diffusion training steps, $N_g$: rehearsal generation interval, $T$: number of tasks, $B$: batch size, $f_\phi$: classifier model, $\epsilon_\theta$: diffusion model
    **for** $i \in [1, \ldots, T]$ **do**
        $\mathcal{D}_i \leftarrow$ real dataset for task $i$
        $\epsilon_{\theta_i} \leftarrow \epsilon_{\theta_{i-1}}$
        $f_{\phi_i} \leftarrow f_{\phi_{i-1}}$
        $\hat{\mathcal{B}} \leftarrow \emptyset$
        **for** $n \in [1, \ldots, N_c]$ **do**                                   ▷ Classifier training
            **if** $i == 1$ **then**
                $\mathcal{B} \leftarrow B$ real samples from $\mathcal{D}_i$
            **else**
                $\mathcal{B} \leftarrow {}^{B}\!/i$ real samples from $\mathcal{D}_i$
                **if** $n \bmod N_g == 0$ **then**
                    $\hat{\mathcal{B}} \leftarrow \texttt{GUIDE\_sample}(\texttt{num\_samples} = {}^{B}\!/i \times (i-1), f_{\phi_i}, \epsilon_{\theta_{i-1}})$
                **end if**
            **end if**
            $\mathcal{B}_c \leftarrow \mathcal{B} \cup \hat{\mathcal{B}}$
            update $f_{\phi_i}$ with $\mathcal{B}_c$
        **end for**

        $\hat{\mathcal{D}}_i \leftarrow \emptyset$                                   ▷ Construct diffusion dataset
        **if** $i > 1$ **then**
            $\hat{\mathcal{D}}_i \leftarrow \texttt{sample\_without\_guidance}(\texttt{num\_samples} = |\mathcal{D}_{1,\ldots,i-1}|, \epsilon_{\theta_{i-1}})$
        **end if**
        $\mathcal{D}_d \leftarrow \mathcal{D}_i \cup \hat{\mathcal{D}}_i$
        train $\epsilon_{\theta_i}$ on $\mathcal{D}_d$ for $N_d$ steps
    **end for**

---

# D    TRAINING DETAILS AND HYPERPARAMETERS

## D.1    DIFFUSION MODELS

In our experiments, we follow the definitions of class-conditional diffusion model architectures described in Dhariwal & Nichol (2021). Across all setups, we train diffusion models using AdamW optimizer with $\beta_1 = 0.9$ and $\beta_2 = 0.999$. The only hyperparameter that varies between tasks is the number of iterations of training. In all setups, we train models for 100K iterations on the initial task, and then on each subsequent task, we train either for 50K or 100K iterations, depending on the setup. We use DDIM with 250 steps for the generation of rehearsal samples on the ImageNet100 dataset and 1000 denoising steps for every other dataset. The only augmentation that we use during the training of diffusion models is the random horizontal flip.

We present the most important hyperparameters for diffusion models in Tab. 8.

## D.2    CLASSIFIERS

As a classifier model, we use the same ResNet18 architecture as the GDumb method (Prabhu et al., 2020) in each setup, with preactivation enabled, meaning that we place the norms and activations before the convolutional or linear layers. We train models with an SGD optimizer. We list the most important hyperparameters for classifiers in Tab. 9.

For CIFAR-10 and CIFAR-100 datasets, we define the same set of image augmentations, which include cropping, rotating, flipping, and erasing. We also apply a transformation to the brightness, contrast, saturation, and hue, followed by the normalization to the [-1, 1] range. During the training, we apply the same augmentations for both rehearsal samples and real samples from the current task.

For the ImageNet100, we do not apply any data augmentations, but we also normalize the images to the range of [-1, 1].

Table 8: Hyperparameters for the training of diffusion models.

|  | CIFAR-10/2 | CIFAR-10/5 | CIFAR-100/5 | CIFAR-100/10 | ImageNet100-64/5 |
|---|---|---|---|---|---|
| Diffusion steps | 1000 | 1000 | 1000 | 1000 | 1000 |
| Noise schedule | linear | linear | linear | linear | linear |
| Channels | 128 | 128 | 128 | 128 | 192 |
| Depth | 3 | 3 | 3 | 3 | 3 |
| Channels multiple | 1, 2, 2, 2 | 1, 2, 2, 2 | 1, 2, 2, 2 | 1, 2, 2, 2 | 1, 2, 3, 4 |
| Heads | 4 | 4 | 4 | 4 | |
| Heads channels | | | | | 64 |
| Attention resolution | 16,8 | 16,8 | 16,8 | 16,8 | 32,16,8 |
| BigGAN up/downsample | ✗ | ✗ | ✗ | ✗ | ✓ |
| Dropout | 0.1 | 0.1 | 0.1 | 0.1 | 0.1 |
| Batch size | 256 | 256 | 256 | 256 | 100 |
| Learning rate | 2e-4 | 2e-4 | 2e-4 | 2e-4 | 1e-4 |
| Iterations 1st task | 100K | 100K | 100K | 100K | 100K |
| Iterations other tasks | - | 50K | 50K | 100K | 50K |
| Self-rehearsal denoising steps | - | 1000 | 1000 | 1000 | DDIM250 |

Table 9: Hyperparameters for the training of classifier models.

|  | CIFAR-10/2 | CIFAR-10/5 | CIFAR-100/5 | CIFAR-100/10 | ImageNet100-64/5 |
|---|---|---|---|---|---|
| Batch size | 256 | 256 | 256 | 256 | 100 |
| Learning rate 1st task | 0.1 | 0.1 | 0.1 | 0.1 | 0.1 |
| Learning rate other tasks | 0.01 | 0.01 | 0.05 | 0.05 | 0.001 |
| Iterations 1st task | 5K | 5K | 10K | 10K | 20K |
| Iterations other tasks | 2K | 2K | 2K | 2K | 20K |
| Rehearsal denoising steps | DDIM50 | DDIM50 | DDIM100 | DDIM100 | DDIM50 |
| Rehearsal generation interval | 1 | 5 | 10 | 10 | 15 |

### D.3 EXPERIMENTS COMPUTE RESOURCES

In this section, we list the hardware we used for our experiments and training times on each benchmark, both for training of diffusion models in Tab. 10 and classifiers in Tab. 11. Although we use NVIDIA A100 GPUs in our experiments due to the efficiency of training, experiments can also be reproduced on GPUs with smaller memory, as mentioned in Appendix F.

The total compute used in a project can be estimated from the times presented in tables, considering that each experiment is calculated with three different random seeds for statistical significance. Furthermore, we need to take into account compute used in the development stage of the project.

Table 10: Training times of diffusion models for each benchmark.

| Benchmark | Time[GPU-hours] | GPU used |
|---|---|---|
| CIFAR-10/2 | 5.55 | 4 x NVIDIA A100 40GB |
| CIFAR-10/5 | 20 | 4 x NVIDIA A100 40GB |
| CIFAR-100/5 | 20 | 4 x NVIDIA A100 40GB |
| CIFAR-100/10 | 63.75 | 4 x NVIDIA A100 40GB |
| ImageNet100-64/5 | 88.43 | 4 x NVIDIA A100 40GB |

Table 11: Training times of classifiers for each benchmark.

| Benchmark | Time[GPU-hours] | GPU used |
|---|---|---|
| CIFAR-10/2 | 4.8 | 1 x NVIDIA A100 40GB |
| CIFAR-10/5 | 5.8 | 1 x NVIDIA A100 40GB |
| CIFAR-100/5 | 5.35 | 1 x NVIDIA A100 40GB |
| CIFAR-100/10 | 13.53 | 1 x NVIDIA A100 40GB |
| ImageNet100-64/5 | 30.73 | 4 x NVIDIA A100 40GB |

# E FORGETTING IN DIFFUSION MODELS

## E.1 COVERAGE OF DATA MANIFOLD

In this analysis, we explore the forgetting behavior of a diffusion model trained continually on CIFAR-10, divided into two tasks with 25000 training samples each. Initially, we train the model exclusively on data from the first task (*Real Task 1*). Subsequently, we employ two approaches: standard self-rehearsal training for the second task (*Continual Task 2*), akin to our method, and retraining the first task's model on the entire CIFAR-10 dataset (*Upper-bound Task 2*), which serves as our upper bound. All training conditions, including architectures, training steps (100K), and hyperparameters, remain consistent across setups. Generative metrics (FID (Heusel et al., 2017), Precision, and Recall (Kynkäänniemi et al., 2019)) for samples generated with DDIM100 from all models are displayed in Tab. 12.

Table 12: Comparison of FID, Recall, and Precision metrics of all three trained models. We calculate metrics for 25000 samples generated using DDIM100. When we train the diffusion model in a continual manner, we observe a very significant drop in the Recall.

|  | FID ($\downarrow$) | RECALL ($\uparrow$) | PRECISION ($\uparrow$) |
|---|---|---|---|
| REAL TASK 1 | 9 | 55 | 65 |
| UPPER-BOUND TASK 2 | **13** | **52** | 64 |
| CONTINUAL TASK 2 | 26 | 37 | **67** |

Continual training of the diffusion model leads to a noticeable reduction in its capability to cover the training manifold of Task 1, as evidenced by a decrease in the Recall metric. However, the Precision metric does not show a significant drop, indicating that the quality of the generated samples remains largely unaffected. Moreover, we visualize the umap embeddings of both real data examples and generated samples in Fig. 8. In plot (c), it's evident that the coverage of real data samples (depicted in blue) by the generated samples has noticeably diminished.

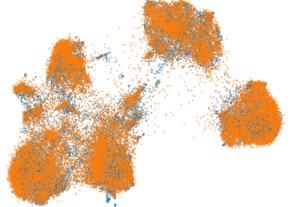 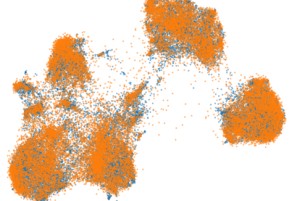 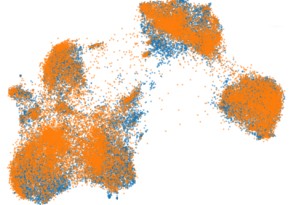

(a) Initial task 1 data manifold.

(b) Task 1 data manifold after training on task 2 – **upper-bound**.

(c) Task 1 data manifold after training on task 2 – **continual training**

Figure 8: **Visualization of task 1 data manifold in the CIFAR-10 dataset.** Blue points represent embedded real data samples from task 1, and orange points represent generated samples. We see a significant drop in coverage of training data manifold after training on the second task of continual training.

## E.2 Forgetting of training data samples

In our study, we further analyze what data samples the diffusion model trained continually tends to forget. We conducted an experiment where we sample examples from class *apple* from our class-conditional diffusion model, trained on the CIFAR-100 dataset divided into 5 tasks, after completing the second task. Utilizing Precision and Recall metrics definitions (Kynkäänniemi et al., 2019), we computed representations using the Inception-v3 model for both real and generated sample sets. This allowed us to identify which segments of the approximated real data manifold were covered or missed by the generated manifold.

Figure 9a illustrates examples of remembered data samples, while Figure 9b depicts those data samples that were forgotten by the diffusion model due to continual training. Consistent with findings by Toneva et al. (2018), our diffusion model tends to forget rare samples in the training

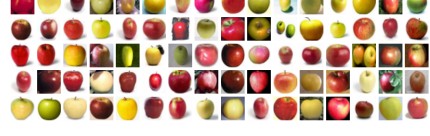

(a) Samples from class *apple* remembered by the diffusion model after training on the second task.

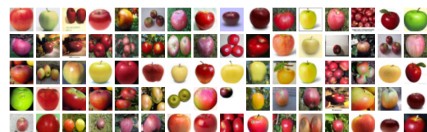

(b) Samples from class *apple* forgotten by the diffusion model after training on the second task.

set, such as those with complex backgrounds, while retaining more common samples characterized by simpler backgrounds and typical shapes. This is also connected to the drop in Recall, where from task to task, the diffusion model loses the ability to generate samples from low-density regions of data manifold (Sehwag et al., 2022).

## F Runtime analysis of GUIDE for CIFAR-10/5

We perform an extensive runtime analysis in order to compare the training times of GUIDE to other existing baselines. For the training of each method, we used the same machine with a single NVIDIA RTX A5000 GPU. Since in the training of GUIDE on CIFAR-10/5 setup, we generate new rehearsal examples every 10 batches, for fairness, we evaluate the DDGR runtime with exactly the same interval. We present the calculated runtimes in Tab. 13.

Table 13: Runtime analysis of all baseline methods on the CIFAR-10/5 benchmark. (*) We evaluate the DDGR method with the same rehearsal generation interval as we use in GUIDE on this benchmark – once every 10 batches.

| Method | Time [GPU-hours] |
|---|---|
| DGR VAE | 1.28 |
| DGR+distill | 1.35 |
| RTF | 1.09 |
| BIR | 2.49 |
| GFR | 1.76 |
| DDGR | 111.96* |
| DGR diffusion | 76.1 |
| **GUIDE** | 82.95 |

Approaches that train diffusion models (DDGR, DGR diffusion, and GUIDE) require significantly more computations. However, GUIDE does not increase much of the runtime compared to the standard DGR with diffusion model ($+9\%$ wall time). At the same time, when comparing GUIDE to the recently proposed DDGR (Gao & Liu, 2023) method that also uses diffusion models in generative replay scenario, our approach needs notably less computation time ($-26\%$ wall time) while achieving better performance in all evaluated setups.

# G    SAMPLES FROM IMAGENET100 128x128

Mushroom

Scuba diver

Strawberry

Pomegranate

Irish setter

Catamaran

Banana

Lemon

Airliner

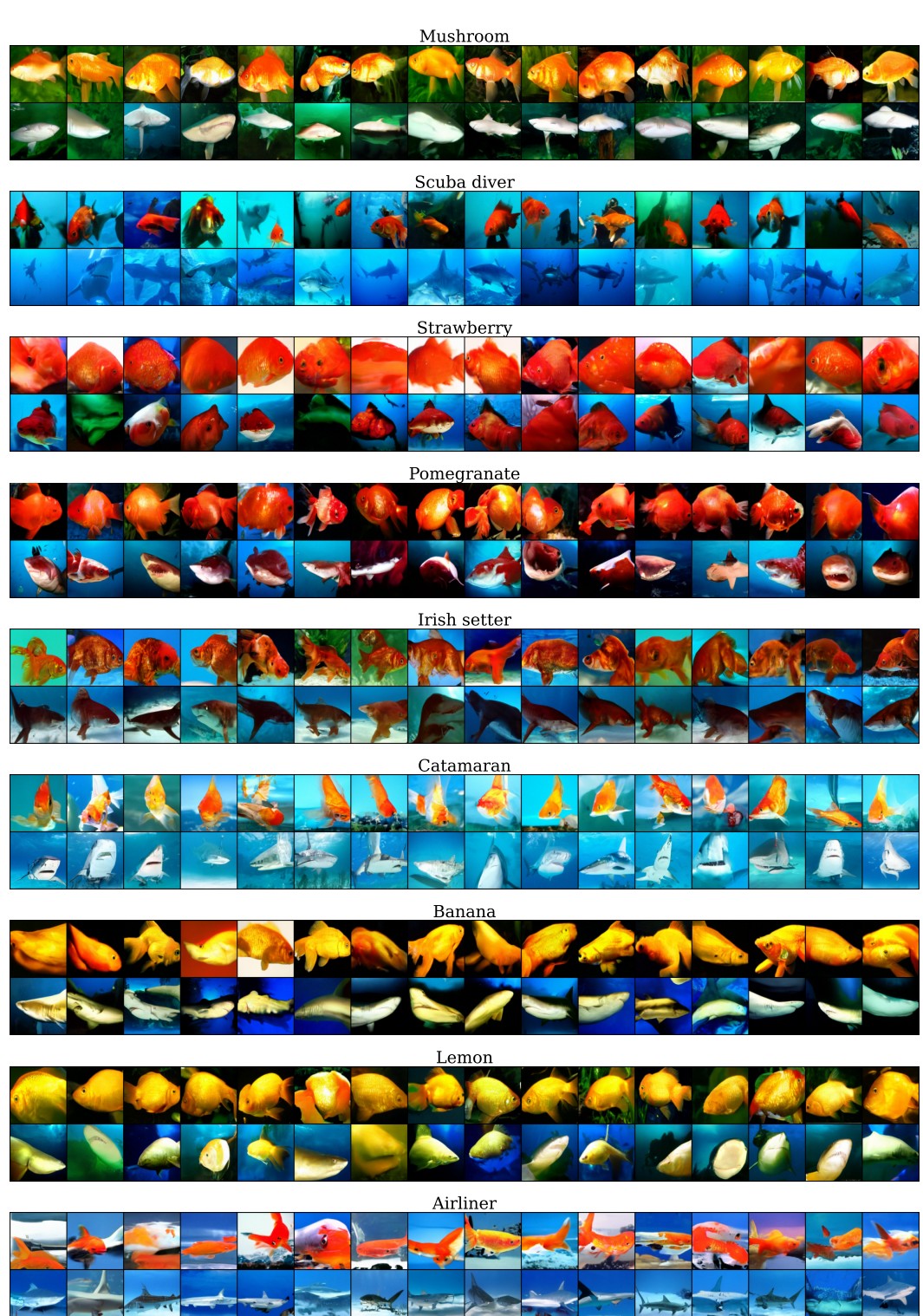

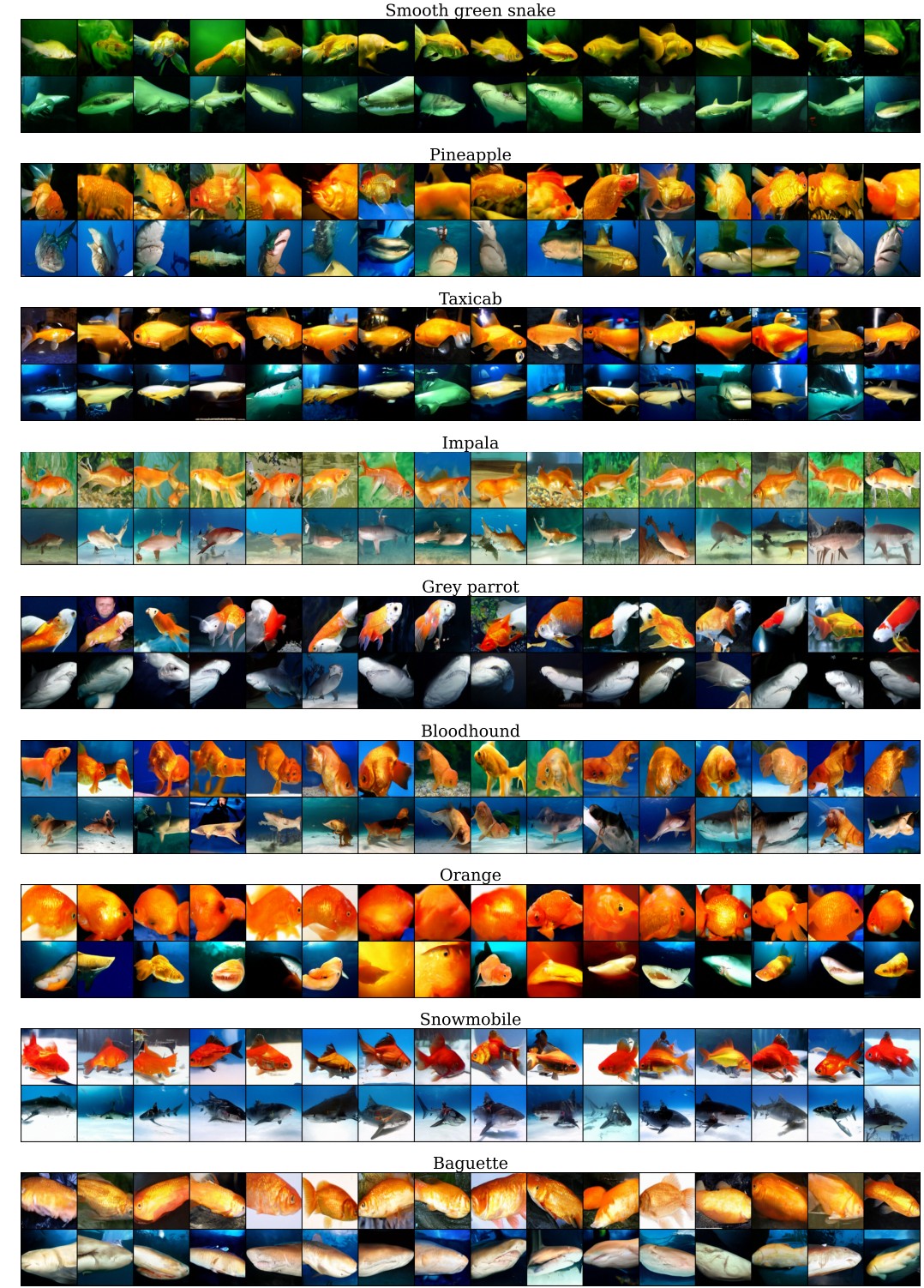

Figure 10: **Random samples generated from an unconditional diffusion model trained only on** *goldfish* **and** *tiger shark* **classes from the ImageNet100 128x128 dataset.** Each image grid presents samples generated with guidance to the class depicted above the grid which is unknown to the diffusion model. We generate samples using 1000 denoising steps and we set guidance scale $s$ to 10 for both classes that we guide to.

## H SAMPLES FROM CIFAR-10

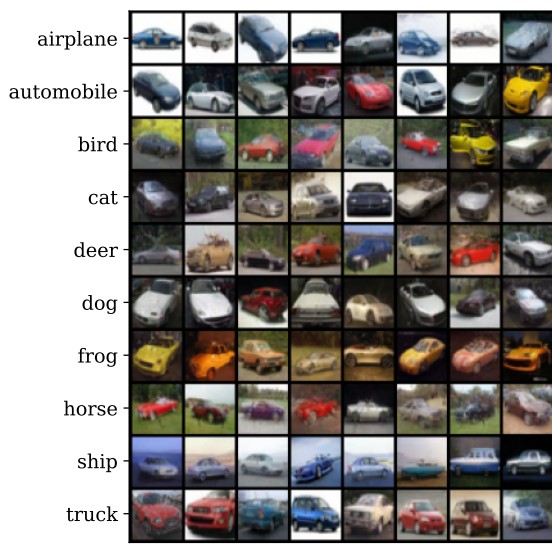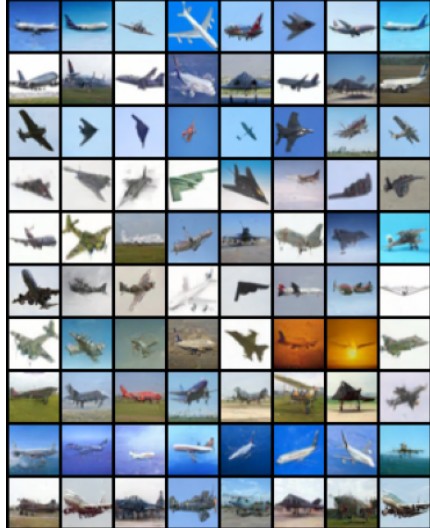

Figure 11: **Samples from unconditional diffusion model trained only on classes *automobile* and *airplane* from the CIFAR-10 dataset.** We generate those samples following the same procedure as discussed in Sec. 3.1. On the left grid, we guide the denoising process to automobiles, while on the right grid, to airplanes. At the same time, we add the guidance to classes depicted on the y-axis of the figure. We generate samples using 1000 denoising steps and we set guidance scale $s$ to 5 for both classes that we guide to.

## I BROADER IMPACTS

This paper presents work whose goal is to advance the field of continual machine learning. There are many potential societal consequences of our work, many of which are generic to the machine learning field in general, i.e., the DGR algorithm considered in this paper will reflect the biases present in the dataset. Hence, it is important to exercise caution when using this technique in applications where dataset biases could lead to unfair outcomes for minority and/or under-represented groups. In our case, this especially concerns training a diffusion model used for replay, in which simple random sampling can result in a different data distribution than the original dataset. It would be worthwhile to actively monitor the model's outputs for fairness or implement bias correction techniques to mitigate these negative impacts.

Additionally, while diffusion models have shown promise in various generative tasks, their adaptability through fine-tuning and continual learning remains relatively unexplored. In this early-stage research, some potential risks can emerge while combining our proposed algorithm of guidance with malicious models or when performing more sophisticated attacks.

