# OpenReview forum: "GUIDE: Guidance-based Incremental Learning with Diffusion Models"
_ICLR.cc/2025/Conference — ICLR 2025 Conference Withdrawn Submission_

### Official Review · Reviewer_TYsP · 2024-11-01

**Soundness:** 3
**Presentation:** 2
**Contribution:** 2
**Rating:** 5
**Confidence:** 5

**Summary:**

The paper studies the problem of task incremental continual learning with generative replay. The paper proposes to use diffusion models as the sample generator and guide the diffusion generation process so that the generated samples are most ''confusing'' with the samples of the current task and therefore those samples are most likely to be forgotten. The paper shows that empirically the proposed method outperforms other generative replay CL approaches.

**Strengths:**

1. The paper introduces the idea of using information from the current task in generating replay samples to generative replay CL.

2. Compared to diffusion-based generative replay CL, the proposed method seems to be a more effective way to utilize the diffusion model.

**Weaknesses:**

1. The technical novelty and contribution is limited. The guidance of diffusion model generation follows standard procedures and the high-level idea follows the memory replay CL approach of RAR.

2. The presentation can be improved. Many figures and tables are extremely small (e.g., table 1 and figure 2). Also, eq 6 is confusing. Based on the definition of y_i in eq 7, the y_{i-1} in eq 6 should only refer to a certain class of task i-1, which I don't think is the intention of the author (correct me if I am wrong). A more reasonable definition of y_{i-1} is given much later in the discussion.

3. This is a general weak point for the generative replay CL approach. Compared to memory replay CL, the generative approach requires costly and potentially offline training of the generative model, which makes the approach less practical in real scenarios. The paper does not push forward in the direction of solving this practical issue.

**Questions:**

1. The diffusion sampling (DDIM with 250 steps or so) is relatively costly considering modern standards (e.g., DDIM with 50 or fewer steps or more advanced samplers like unipc with 10-20 steps). How does the performance change if we aim to reduce the sampling costs? Also, if sampling is cheaper, we can sample more hard training data points. How does the performance change then?

2. Is the diffusion model trained progressively for the main result? I.e., first train the model on task 1 and then train on combined task 1 and task 2, and so on. Also, what's the performance of the diffusion model trained on the generated data (I think the main results all use real data for the diffusion model)?

---

### Official Review · Reviewer_GkNt · 2024-11-02

**Soundness:** 2
**Presentation:** 3
**Contribution:** 1
**Rating:** 5
**Confidence:** 4

**Summary:**

This paper focuses on catastrophic forgetting and proposes a generative based continual learning method called GUIDE which incorporates classifier guidance into the diffusion process to produce rehearsal examples specifically targeting information forgotten by a continuously trained model. Experiments are provided to empirically evaluate the proposed method.

**Strengths:**

1. This paper is well-written and easy to follow.

2. I appreciate the extensive experiments.

**Weaknesses:**

My main concern is about the technique contribution of this paper.

1. The main technique of this paper is eq.1 and eq.2,  which comes from [1]. In the view of technique, this paper is not novel. This means that this paper is an (a+b)-like work which merges the technique in [1], diffusion model and continual learning setting. Therefore, i think the lack of technique contribution is the main weakness of this paper.

2.  (a+b)-like work is ok, but this paper doesn't show enough motivations about why the proposed method is a proper solution. The motivation needs to described in details.


[1] Arpit Bansal, Hong-Min Chu, Avi Schwarzschild, Soumyadip Sengupta, Micah Goldblum, Jonas Geiping, and Tom Goldstein. Universal guidance for diffusion models. In Proceedings of the IEEE/CVF Conference on Computer Vision and Pattern Recognition, pp. 843–852, 2023.

**Questions:**

1. In line 96, $y$ in $f _{\phi}\left(y \mid \hat{\mathbf{z}} _{0}\left(\mathbf{x} _{t}\right)\right)$ seems to be useless. Should $f _{\phi}\left(y \mid \hat{\mathbf{z}} _{0}\left(\mathbf{x} _{t}\right)\right)$ be replaced by $f _{\phi}\left(\hat{\mathbf{z}} _{0}\left(\mathbf{x} _{t}\right)\right)$?

2. In line 203, it is stated that "$\mathcal{M} _{k} \subset \mathcal{D} _{k}$ are free parameters". Is $\mathcal{M} _{k}$ not the memory buffer  of samples corresponding to task $k$?

3. $\mathcal{M} _{k} $ is generated according to eq.6 which considers the guided information according to the class of current task $i$ but ignores task $k$. So how does it make sure that the generated samples buffer $\mathcal{M} _{k}$ corresponds to previous task $k$?

4. The contexts from lines 235 to 239 seem to demonstrate that the ways in Section 3.2 is reasonable, but it cannot explain Q3.

5. In line 219, it is stated that "... classes, in each denoising step $t$, ....". Does this mean that the selected class $y _i$ can be different in each step $t$ of the sample generating process? If so, is it reasonable?

6. What diffusion models are this paper based on, DDPM, DDIM or something else?

---

### Official Review · Reviewer_U4g5 · 2024-11-03

**Soundness:** 3
**Presentation:** 3
**Contribution:** 3
**Rating:** 5
**Confidence:** 3

**Summary:**

This paper tackles the task incremental continual learning with the help of generative replay. To do so, they use the diffusion model to generate examples from the previous class, that look similar to the examples of the current class. This is achieved by using classifier guidance. Hypothesis is that examples from the previous tasks that are similar to the current one are most prone to forgetting. Using the method mentioned above, they compare their technique GUIDE against different generative replay based methods, outperforming the baselines. Ablations and sensitivity for the sampling hyperparameters are done to show the robustness of the proposed approach.

**Strengths:**

The paper provide good intuition of their approach, showcasing it using different qualitative and 2D projection based examples, as to how different techniques sample the space. I also like the hyperparameter sensitivity performed in order to show robustness, and different possible variant of GUIDE. While I've not been in touch with Continual learning community for a while, I think these bare minimum things are necessary.

**Weaknesses:**

In terms of weakness, I've a few questions

- The sampling procedure as outlined in Alg2 in the appendix uses the model $f_{\phi}$ which may or may not be trained well enough in order to be able to understand which class is the image from the previous task  (as generated by the diffusion model during the sampling steps) belong to in the current set of classes in the task. Moreover, shouldn't it affect the classifier guidance as well when sampling from the previous task's class? This conceptually confuses me.
- I think an ideal situation would have been where one keeps two classifiers, using the one from the immediately previous task form the classifier guidance sampling, and the other (which is being learned for the current task) for providing the gradient of the CE loss.

On computation, and memory footprint.

- How large is the final Diffusion model in terms of size? I feel like in practice if the diffusion model costs (space wise) more than storing important (of not all) set of images from the current task, then it defeats the purpose of using the diffusion model.
- The amount of training time for the diffusion model (as mentioned in the appendix) for even a 100 class dataset like Imagenet 100 seems large to me.

On loss terms and Baselines

- What is the overall loss used in this framework? DER++ outperforms GUIDE at certain places, but couldn't one use DER++ loss for this work?
- Can the authors compare against RAR? (Kumari et al. 2022)

**Questions:**

Refer to the weakness.

---

### Official Review · Reviewer_1Eqd · 2024-11-04

**Soundness:** 3
**Presentation:** 2
**Contribution:** 3
**Rating:** 3
**Confidence:** 4

**Summary:**

The authors propose a deep generative replay methodology for continual learning, which leverages a diffusion model to generate samples that are rehearsed to reduce forgetting. The main idea resides in using *classifier guidance* to guide the generations towards examples that are close to the decision boundary. These samples should be more effective when rehearsed, as they are generated to represent the separation between real examples of different tasks. The methodology proves to be effective on standard continual learning benchmarks.

**Strengths:**

1) Generating examples in a specific way (i.e., on the decision boundary) to relieve forgetting is novel and compelling
2) leveraging the past classifier as a guidance offers interesting insights
3) the ablation studies outline the impact of different components effectively

**Weaknesses:**

1) Experimental results:
 - I did not find the experimental setting clear (e.g., lacking the number of epochs/iterations for each setting)
 - Results reported in Table 1 are suspiciously low (w.r.t. to the results reported by competitors such as DDGR). For instance, DDGR reports an accuracy that is almost **four** times higher on CIFAR-100, 10 tasks.
 - Results in Table 2 are misleading. All methodologies reported as competitors (except for GSS) were introduced for a multi-epoch setup: evaluating them by devising a single epoch may lead to inconsistent and unfair analyses. Certain techniques, such as DER, DER++, DGR Diffusion, and the proposed GUIDE, surpass the continual-joint upper bound in various datasets, highlighting fundamental issues within the current setting (e.g., underfitting of the dataset). Moreover, the same competitor (DER, DER++, iCaRL etc.) yield much higher performance in the original experimental settings, which highlights the need for more training epochs or higher learning rate.
2) Clarity: the explanation of the methodology can generally be improved:
 - A specific aspect is found in line 203, page 4. Here, M_k and D_k are referred to as free parameters, which is inconsistent with their previous definition (D is the dataset at line 196).
 - At first (line 160) the authors introduce an unconditional diffusion model trained exclusively on the goldfish and tiger shark classes from the ImageNet100 dataset. It seems like the diffusion model used for generating the data is always pre-trained, but this is not the case as showed in the algorithms (appendix).
 - Line 168, equation 3. If we wanted to guide the diffusion process away from a class and towards another one, I would expect the two correction factors to have different signs, reflecting opposing directions in gradient space. However, both correction factors currently point in the same direction, which contradicts this expectation.
 - Line 219: what does it mean "a current task"? is there more than one current task? Then, at line 223. The expression "current task i" is misleading: is it the current task or the i^th task?
 - Table 1, 3 and 4 are very small and difficult to read.
 - Line 445: why should maximizing *entropy* of the classifier be of help? Maybe, the authors meant *cross-entropy*?
Additionally, in the caption of Figure 6, the claim of "approaching the continual joint training" is not really met, as there still is a difference of 10+ percentage points.

If find the main idea of this work original and interesting, but I think that its application lacks clarity and experimental significance.

**Questions:**

None

---

### Note · Authors · 2024-11-22

**Comment:**

We sincerely thank the Reviewers for their time and valuable feedback. After careful consideration, we have decided to withdraw this submission.

**Withdrawal Confirmation:**

I have read and agree with the venue's withdrawal policy on behalf of myself and my co-authors.